# Buckling and Post-Buckling Behavior of Perfect/Perforated Composite Cylindrical Shells under Hydrostatic Pressure

**Ke-Chun Shen [1,2,3,*], Zhao-Qi Yang [1,2], Lei-Lei Jiang [1,2] and Guang Pan [1,2]**

1   School of Marine Science and Technology, Northwestern Polytechnical University, Xi'an 710072, China;
    yzq060615@163.com (Z.-Q.Y.); 18192391636@mail.nwpu.edu.cn (L.-L.J.); panguang@nwpu.edu.cn (G.P.)
2   Key Laboratory for Unmanned Underwater Vehicle, Northwestern Polytechnical University,
    Xi'an 710072, China
3   Structural Ceramics and Composites Engineering Research Center, Shanghai Institute of Ceramics,
    Chinese Academy of Sciences, Shanghai 200050, China
*   Correspondence: shenkechun@126.com

**Abstract:** In this paper, the buckling and post-buckling behavior of perfect and perforated composite cylindrical shells subjected to external hydrostatic pressure was experimentally investigated. Three filament wound composite cylindrical shells were fabricated from T700-12K Carbon fiber/Epoxy, two of which were perforated and reinforced. A test platform was established that allows researchers to observe the deformation of composite cylindrical shells under hydrostatic pressure in real-time during test. According to experimental observation, strain response and buckling deformation wave were discussed. Comparative analysis was carried out based on the experimental observation and finite element prediction. Results show that the deformation of composite cylindrical shell under hydrostatic pressure included linear compression, buckling and post-buckling processes. The buckling behavior was a progressive evolution process which accounted for 20% of the load history, and strain reversal phenomenon generally occurred at the trough of the buckling wave. As for the postbuckling deformation, the load carrying capacity of the shell gradually decreased while the magnitude of strain continued increasing. Both the perfect and perforated composite cylindrical shells collapsed at the trough of the buckling wave. Comparing with the perfect shell, it was validated the reinforcement design could effectively ensure the load carrying capacity of the perforated composite cylindrical shell.

**Keywords:** composite cylindrical shell; critical buckling pressure; mode; hydrostatic pressure

## 1. Introduction

Composite materials have excellent mechanical properties such as specific strength and stiffness, low density and corrosion resistance, and have been widely used in marine engineering fields [1]. Composite cylindrical shell is a typical structure to resist high external pressure and has been increasingly applied in underwater vehicles [2].

Buckling [3–7] is a major concern for scholars when designing thin-walled composite cylindrical shells. Formulas for the calculation of critical buckling pressure for composite cylindrical shells have been studied and discussed by many researchers. Lopatin [8,9] proposed an effective analytical solution to predict the critical buckling pressure of composite cylindrical shell subjected to hydrostatic pressure, where the vibration mode shape of clamped-clamped beam was selected as an approximation function and the Galerkin method was used to solve the governing equations. Cho [10] proposed an empirical formula to calculate the critical buckling pressure of composite cylindrical shell considering material failure. Imran [11] carried out a large number of comparative investigations on the elastic buckling of composite cylindrical shells under hydrostatic pressure with various geometric sizes and layup configurations. Based on first-order shear deformation theory and classical laminated plate theory, Ehsani [12] established the numerical model of laminated composite

grid structures and obtained the elastic buckling load by using the Ritz method. Cai [13] presented a design procedure and calculated the buckling pressure of composite pressure shells subjected to external hydrostatic pressure based on the classical laminate theory. With the improvement of computer performance, finite element analysis has become an effective method to evaluate buckling failure of composite structures [14–17]. Moham-mad [18] employed a triangular shell element having six nodes to investigate the nonlinear behavior of cylindrical shells with irregular mesh and complicated geometry, where total Lagrangian formulation was utilized considering the large deflection and rotation. Further, Mohammad [19] developed a three-node triangular shell element using a mixed strain finite element approach for nonlinear analysis of FG shells with large deformations and finite rotations. For composite cylindrical shells of given geometry, boundary conditions and material system, the buckling strength is mainly affected by the fiber orientation, stacking sequence and thermal environment [20–25]. The researchers conducted extensive studies to maximize the buckling strength of composite cylindrical shells under hydrostatic pressure by using optimization algorithms coupled with analytical solution [26,27] or finite element method [28].

External hydrostatic tests helped scholars to understand the load-carrying capacity of the shell structure. Some researchers investigated the buckling behavior of composite cylindrical shell subjected to hydrostatic pressure by testing. Carvelli [3] observed considerable deformation and captured collapse instant of the Glass fiber reinforced polymer cylindrical part of an underwater vehicle in off-shore test. Hur [29] investigated the buckling and post-buckling behavior of composite laminated cylinders subjected to hydrostatic pressure and presented the final buckling shape of the composite cylindrical shell. Ross [30] conducted the collapse of 44 circular cylindrical composite tubes under hydrostatic pressure. Moon [31] investigated the buckling and failure characteristics of composite cylindrical shells with winding sequences of $[\pm 30/90]_{FW}$, $[\pm 45/90]_{FW}$ and $[\pm 60/90]_{FW}$ under external hydrostatic pressure, where the strain response indicated that nonlinear behavior occurred before the shell collapsed. The researchers defined the collapse pressure as the critical buckling pressure and did not clearly explain the nonlinear strain behavior. In the previous study [32], the author distinguished the difference between buckling pressure and collapse pressure of composite cylindrical shell under hydrostatic pressure from the test; however, the buckling evolution process was not investigated as the shell was closed at both ends and the deformation image cannot be acquired.

From the review of published literature studies, it is found that the researchers generally obtained the critical buckling pressure directly from the collapse pressure in test, analytical solution and numerical method. There is a lack of research on the study of the buckling evolution process, and in most cases, the shell is closed at both ends, making it difficult to observe the buckling shape in test. On the other hand, vertical or lateral thrusters could be equipped on the shell structure to improve the maneuverability of underwater vehicles. Therefore, it is inevitably to open perforated holes on the cylindrical shell, which would cause the loading carrying capacity of the structure to change. Focus on the buckling evolution process of perfect and perforated composite cylindrical shell. In the present work, three filament wound composite cylindrical shells were manufactured, and experimental studies were conducted. A test platform by integrated strain instrument, high-speed camera and hydrostatic chamber was established, allowing researchers to monitor the buckling evolution of the composite cylindrical shell under hydrostatic pressure in real-time. Strain response and buckling behavior were discussed. Some new finds were concluded, which could provide guidance for the safety design of composite cylindrical shell structure for underwater vehicle applications.

## 2. Experiment Design

The external hydrostatic pressure test of the deep-sea shell structure was mostly carried out in a closed hydrostatic chamber. Thus, it was difficult to observe the buckling behavior and evolution characteristics. For this reason, a test platform suitable for real-time

observation shell deformation under hydrostatic pressure was built. As shown in Figure 1, the test platform consisted of a hydrostatic chamber, a high-speed camera and a strain acquisition instrument. The composite cylindrical shell was connected to the flange to form a closed space with the hydrostatic chamber. The high-speed camera was set up above the chamber, and the strain gauges were pasted in the inner wall of the composite cylindrical shell and connected to the acquisition instrument through wires. A high-pressure water pump was used to continuously inject water into the hydrostatic chamber through the injection tube. Strain data and shell deformation graphic are simultaneously transmitted to the control computers, allowing the authors to monitor the buckling evolution of the composite cylindrical shell under hydrostatic pressure.

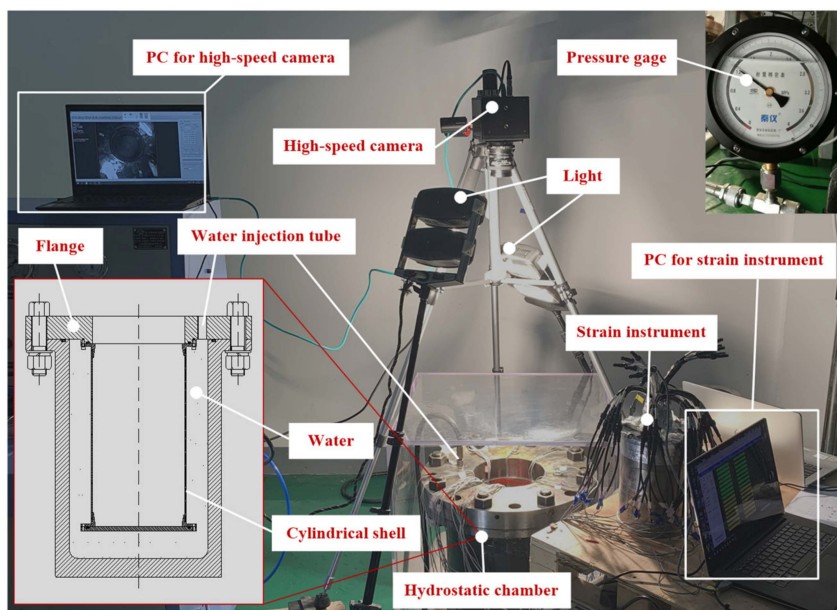

**Figure 1.** Test platform.

The composite cylinder was fabricated by the filament winding process with $[\pm90]_2/([\pm20]/[\pm90]/[\pm40]/[\pm90]/[\pm60]/[\pm90])_2/[\pm90]$ sequence using the T700-12K Carbon fiber/Epoxy composite material, and the cylinder was formed on the core mold. After curing and demolding, it was cut into three cylinders with equal length 375 mm. As shown in Figure 2a, the left control sleeve and right control sleeve, made of aluminum alloy (Young modulus E = 71 GPa, Poisson's ratio v = 0.33), were, respectively, nested on the two ends of each composite cylinder to support the shell. The end cover was attached to the right control sleeve. For a comparative study of the buckling behavior of perfect and perforated composite cylindrical shells, a hole with a diameter of 100 mm was cut in the two of the specimens (see Figure 2b,c). Design of reinforcement was carried out to compensate for the load-carrying capacity loss caused by the hole. Shown in Figure 2d, the reinforcement parts which were made of aluminum alloy consisted of an inner sleeve, an outer sleeve and a jam nut. The inner sleeve and the outer sleeve clamped the edge of the hole, and the jam nut produced a certain pre-tightening force so that the inner sleeve, the outer sleeve and the composite material cylinder were firmly connected. O-rings were used for sealing between inner sleeve and outer sleeve, and sealing between inner sleeve and end cap. The specimens ready for test were marked A#, B# and C# (see Figure 2e). The dimensions of composite cylindrical shell were as follows: the inner diameter 200 mm, length 405 mm and wall thickness 3 mm. The mechanical properties of T700-12K Carbon fiber/Epoxy are listed in Table 1.

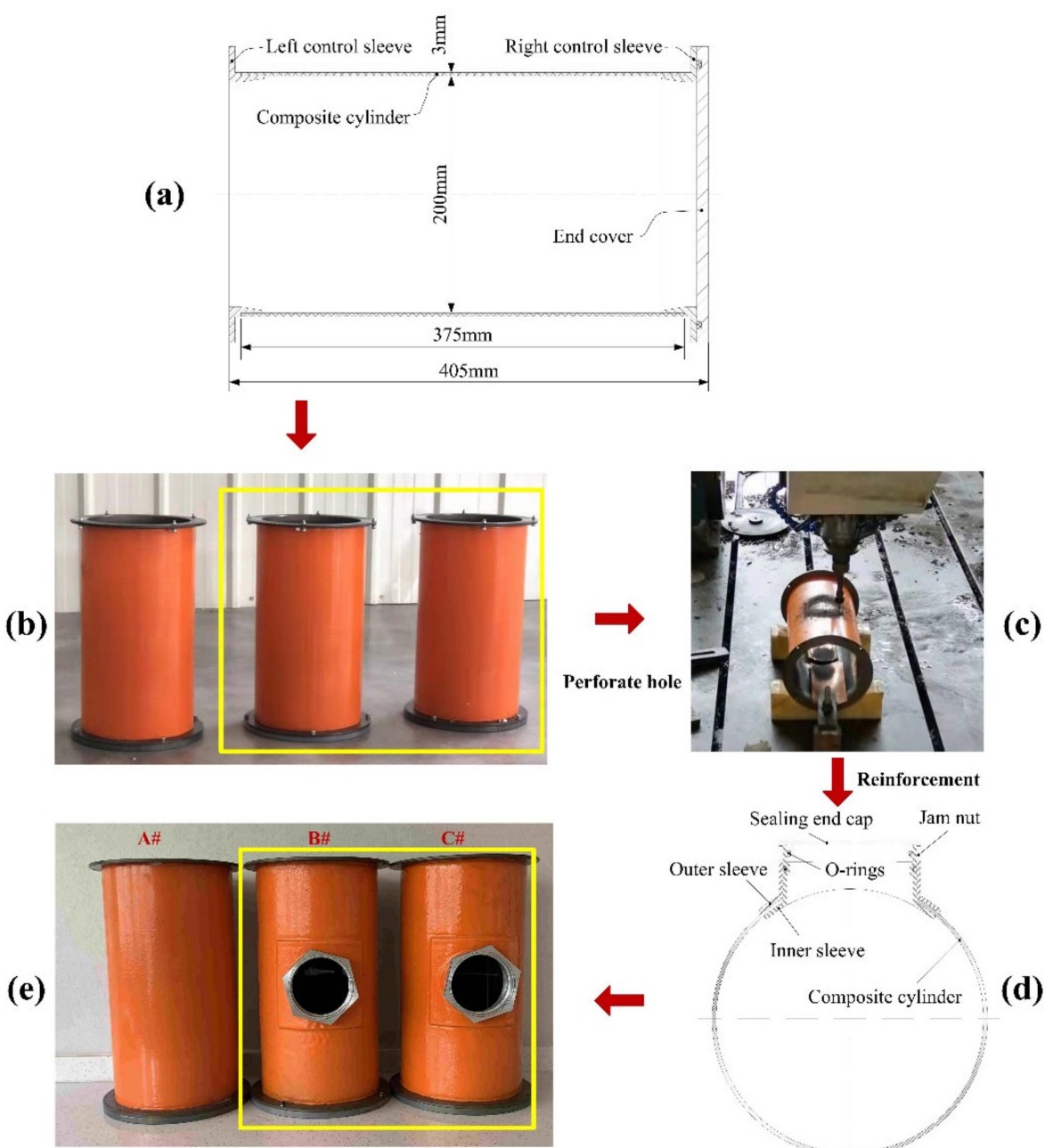

**Figure 2.** Manufacturing process of the composite cylindrical shells for test.

**Table 1.** Properties of T700-12K Carbon fiber/Epoxy.

| Properties | Symbol | Value | Unit |
|---|---|---|---|
| Elastic modulus | $E_{11}$ | 102 | GPa |
| | $E_{22}$ | 7 | GPa |
| | $E_{33}$ | 7 | GPa |
| Poisson's ratio | $v_{12}$ | 0.16 | |
| | $v_{13}$ | 0.16 | |
| | $v_{23}$ | 0.32 | |
| Shear modulus | $G_{12}$ | 8 | GPa |
| | $G_{13}$ | 8 | GPa |
| | $G_{23}$ | 4.5 | GPa |

## 3. Results and Discussion

Three composite cylindrical shells were tested in sequence, and the typical pressure vs. time curves were obtained. Figures 3 and 4 show the evolution of the load-carrying capacity of the perfect composite cylindrical shell A# and the perforated composite cylindrical shell C#, respectively. The bearing capacity curves presented two characteristics: on the one hand, the curves could be divided into two sections, which are monotonically increasing section and monotonically decreasing section; on the other hand, the curves could also be divided into linear section and nonlinear section. According to the linearity, nonlinearity and monotonicity of the curve, the deformation of the composite cylindrical shell consisted of three stages: linear compression, buckling and post-buckling. The starting point of the nonlinear part was defined as initial buckling load $p_{initial}$, the extreme point and end point were defined as terminated buckling load $p_{terminate}$ and failure load $p_{failure}$, respectively.

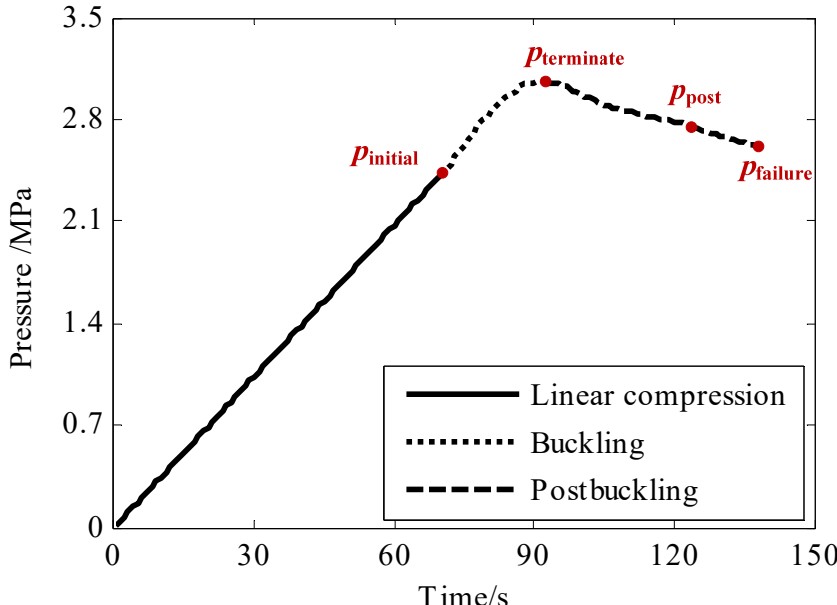

**Figure 3.** Typical pressure vs. time curve of composite cylindrical shell A#.

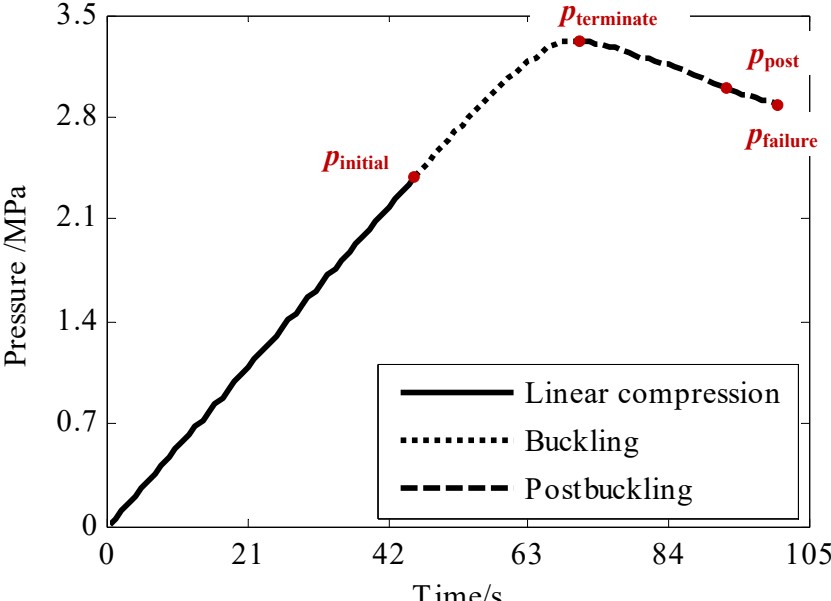

**Figure 4.** Typical pressure vs. time curve of composite cylindrical shell C#.

Table 2 listed the values of initial buckling load, terminated buckling load and failure load. Compared with the perfect composite cylindrical shell A#, the initial buckling pressure of perforated composite cylindrical shell B# decreased by 7.4% and shell C# increased by 0.83%. As for the terminated buckling load and failure load, the perforated composite cylindrical shells B# and C# performed much better. It could be seen that the reinforcement effectively ensured the load carrying capacity of the perforated shell. Observing the buckling shape listed in Table 2, it was found that the buckling shape of the perfect composite cylindrical shell was evenly distributed symmetrically along the circumference. With regard to the perforated composite cylindrical shell, the buckling shape presented non-uniform and asymmetric distribution, which was mainly caused by the non-uniform circumferential stiffness. Specifically, the stiffness of the reinforcement part was greater than that of the surrounding area.

**Table 2.** Experimental results.

| Specimens | A# | B# | C# |
|---|---|---|---|
| $p_{initial}$/MPa | 2.42 | 2.24 (−7.4%) | 2.44 (0.83%) |
| $p_{terminate}$/MPa | 3.06 | 3.09 (0.98%) | 3.32 (8.5%) |
| $p_{failure}$/MPa | 2.58 | 2.64 (2.3%) | 2.88 (11.6%) |
| Deformation at $p_{post}$ |  $p_{post}$ = 2.7 MPa |  $p_{post}$ = 2.74 MPa |  $p_{post}$ = 2.96 MPa |
| Failure morphology |  |  |  |

White silicone rubber was glued on the inner wall of the composite cylindrical shell, and a perfect circle was marked in the images captured by the high-speed camera to represent the outline of the undeformed shell. Comparing the white silicone rubber with the perfect circle, the deformation wave would be identified once the buckling behavior occurred. The deformation shape of the perfect composite cylindrical shell A# was shown in Figure 5. At the moment $p_{terminate}$ = 3.06 MPa, two crests and two troughs were clearly visible. Subsequently, the deformation developed into post-buckling stage where the initial buckling wave was extended widely (in case of $p_{post}$ = 2.7 MPa), and the third trough and the third crest were created gradually. In the post-buckling stage the load carrying capacity decreased gradually. Eventually, the shell collapsed at $p_{failure}$= 2.58 MPa. At the moment of implosion, the cylindrical shell almost recovered its original shape except for a bulge at the blasting position (failure morphology shown in Table 3).

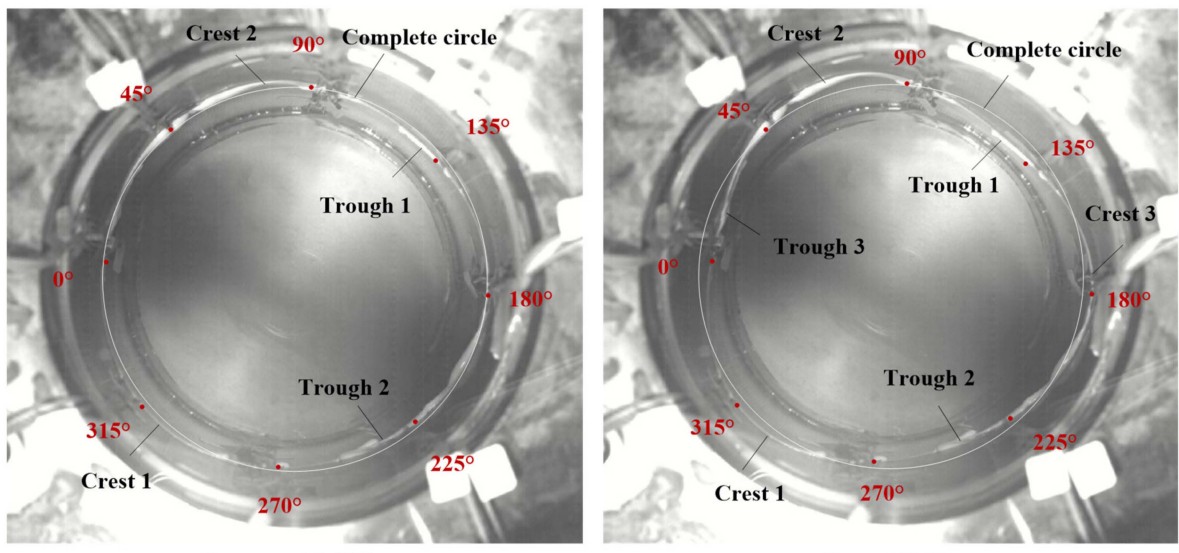

$P_{\text{terminate}}=3.06\text{MPa}$        $P_{\text{post}}=2.7\text{MPa}$

**Figure 5.** Strain gauge locations and deformation shape of the perfect composite cylinder A#.

**Table 3.** Load range at the buckling stage.

| Specimens | A# | B# | C# |
|---|---|---|---|
| Load history/MPa | 0–3.06 | 0–3.09 | 0–3.32 |
| Buckling Stage/MPa | 2.42–3.06 | 2.24–3.09 | 2.44–3.32 |
| Proportion | 20.9% | 27.5% | 26.5% |

As for the perforated composite cylindrical shell C#, the deformation shape was shown in Figure 6. At the moment $p_{\text{terminate}}$ = 3.32 MPa, the buckling wave was faintly visible locally. In the post-buckling stage (in case of $p_{\text{post}}$ = 2.96 MPa), the deformation shape had a relatively large expansion. In the reinforcement part, the width of wave crest was much larger. This phenomenon was caused by the increased stiffness of the reinforcement part. Ultimately, the perforated composite cylindrical shells B# and C# imploded at $p_{\text{failure}}$ = 2.64 MPa and $p_{\text{failure}}$ = 2.88 MPa, respectively. Each of the shell created an axial crack which located by the side of the reinforcement part (shown in Table 3).

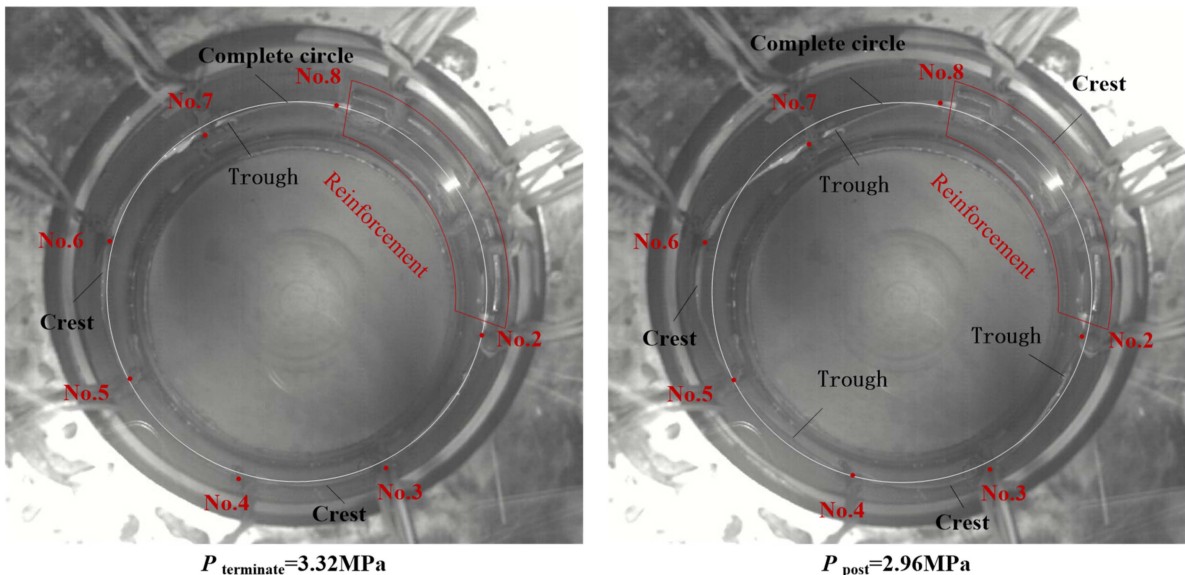

$P_{\text{terminate}}=3.32\text{MPa}$        $P_{\text{post}}=2.96\text{MPa}$

**Figure 6.** Strain gauge locations and deformation shape of the perforated composite cylinder C#.



The proportion of the buckling range to the load history was listed in Table 3. For the perfect composite cylindrical shell A#, it was 20.9%, and for the perforated composite cylindrical shells B# and C#, the proportion was 27.5% and 26.5%, respectively.

The finite element analysis was carried out to predict the buckling pressure and buckling mode of composite cylindrical shell subjected to hydrostatic pressure. Both static and eigenvalue analysis were conducted in sequence. Figure 7 shows the geometric model and finite element model of the composite cylindrical shell. The composite cylinder was meshed using a three-dimensional structural shell element (SHELL 281), which is defined by eight nodes with six degrees of freedom at each node: translations in the *x*-, *y*- and *z*-axes, and rotations about the *x*-, *y*- and *z*-axes. The control sleeves, reinforcement part and end cover were meshed using a three-dimensional tetrahedral structural solid element (SOLID187), which has a quadratic displacement behavior and is well suited to modeling irregular meshes. The element is defined by ten nodes having three degrees of freedom at each node: translations in the *x*-, *y*- and *z*-axes. The contact elements TARGE170 and CONTA174 were used to bond the composite cylinder with control sleeve and reinforcement part together. As for the boundary conditions, the left control sleeve was fixed, and the circumference and the axis direction of composite cylindrical shell met the actual experimental condition. Both axial pressure and external pressure were applied on the surface of the shell (see Figure 7b).

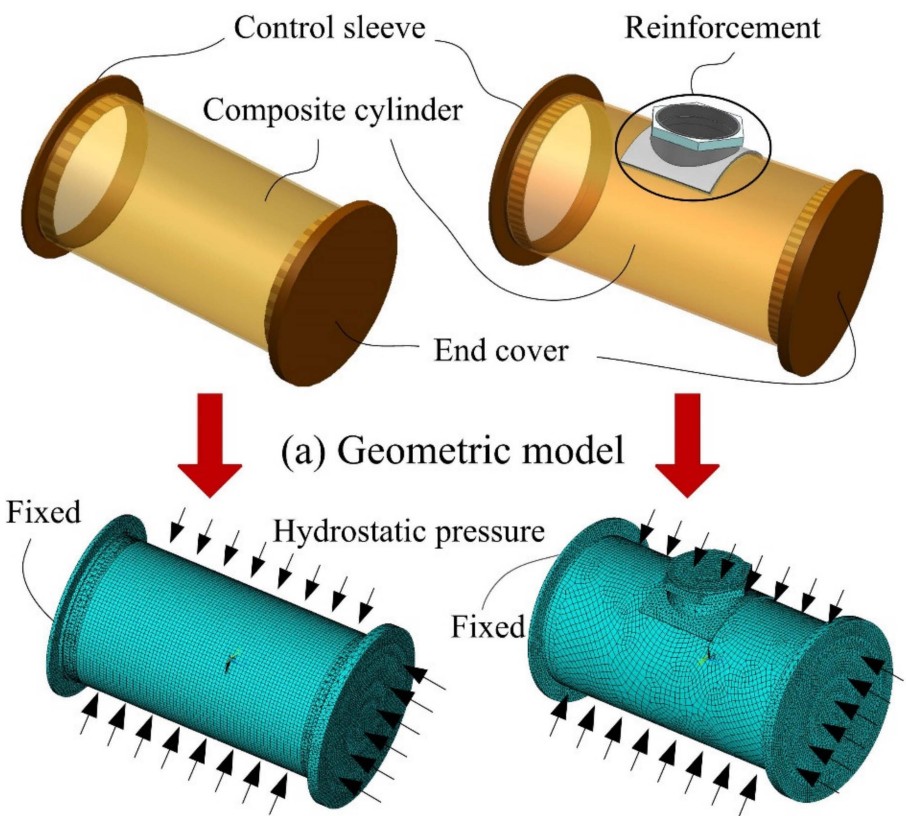

**Figure 7.** Geometric model and finite element model of composite cylindrical shell.

The calculation results were listed in Table 4. It can be noticed that the presence of the reinforced hole did not significantly weaken the load carrying capacity of the composite cylindrical shell, but it slightly improved the load carrying capacity. Comparing the finite element results and the experimental results, it was found that the critical buckling load predicted by FEM was closer to the terminated buckling load obtained by test. As for the

buckling mode, numerical results were in good agreement with the experimental results. The deviations between FE results and experimental terminated buckling load were −1.31%, −0.65% and −7.53%, respectively, which proved that the eigenvalue buckling analysis has the ability to predict the maximum load carrying capacity of the shell accurately.

**Table 4.** Calculation results by FEM.

| Specimens | A# | B# | C# |
|---|---|---|---|
| Critical buckling pressure/MPa | 3.02 | 3.07 | 3.07 |
| Buckling mode |  |  |  |

Circumferential strain gauges (eight gages for perfect cylinder and seven gages for perforated cylinder) were glued on the inner wall of the composite cylindrical shell at the mid-length, and they distributed uniformly every 45° in circumferential direction. In the following strain response analysis, these strain gauges were classified into three groups based on their position such as the crest, trough and boundary sections on the buckling wave (see Table 5). Strain responses of the perfect cylinder A# and perforated cylinder C# are shown in Figures 8–10 and Figures 11–13 respectively. All these strain response curves consisted of three segments which represented the linear compression, buckling and post-buckling deformation stages, respectively. As for the perfect cylinder A#, in the linear compression stage 0–2.42 MPa, the trends in the three group of strain response were extremely similar. Circumferential strain presented compressive and increased linearly with the hydrostatic pressure. In buckling stage 2.42–3.06 MPa, strain magnitude tended to be nonlinearly varying with load. In this process, it was interesting that strain magnitude of the measuring points (0° and 135°) located at the trough of buckling wave gradually changed from negative to positive (see Figure 9). This phenomenon was called strain reversal which was caused by large radial displacement towards the axis of the composite cylinder according to the deformation shape shown in Figure 5. In the post-buckling stage 3.06–2.58 MPa, the load carrying capacity of the composite cylindrical shell gradually decreased while the deformation of the shell became more apparent shown in Figure 5. Therefore, it is easy to explain that the magnitude of positive strain at the trough (in Figure 9) and compression strain at the crest (in Figure 8) continued to increase in postbuckling stage. Observe strain response at the boundary of crest and trough of the perfect cylinder A#. It is strange that strain gages located at 45° and 270° presented compression in the whole load history while Strain gages located at 225° and 90° exhibited reversal phenomenon in the buckling stage.

**Table 5.** Measuring point location.

| Specimens | Crest Section | Trough Section | Boundary of Crest and Trough |
|---|---|---|---|
| A# | 180°, 315° | 0°, 135° | 45°, 90°, 270° 225° |
| C# | No.3, No.6 | No.2, No.7 | No.4, No.5, No.8 |

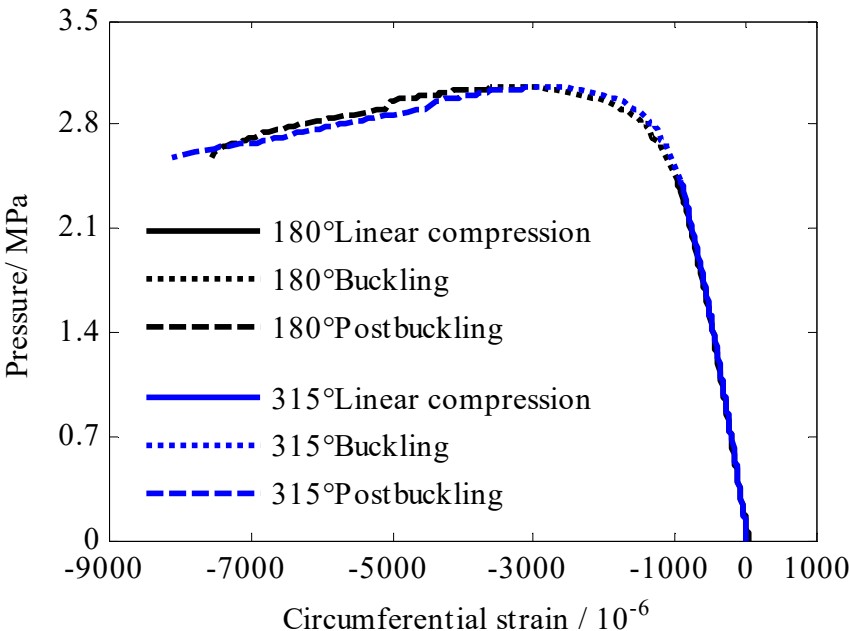

**Figure 8.** Circumferential strain at the crest of the buckling wave of shell A#.

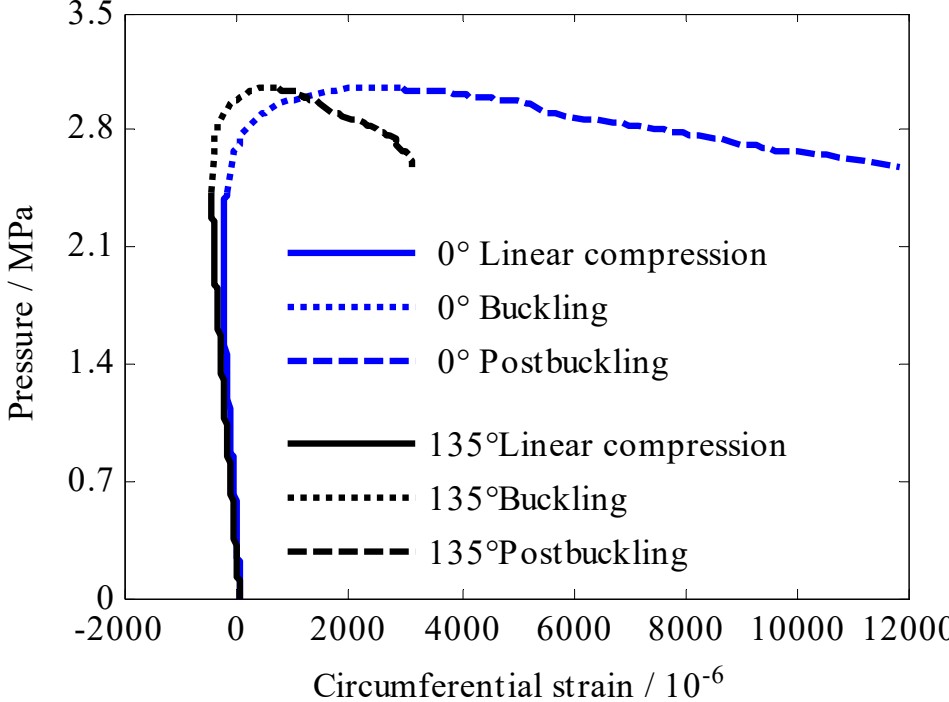

**Figure 9.** Circumferential strain at the trough of the buckling wave of shell A#.

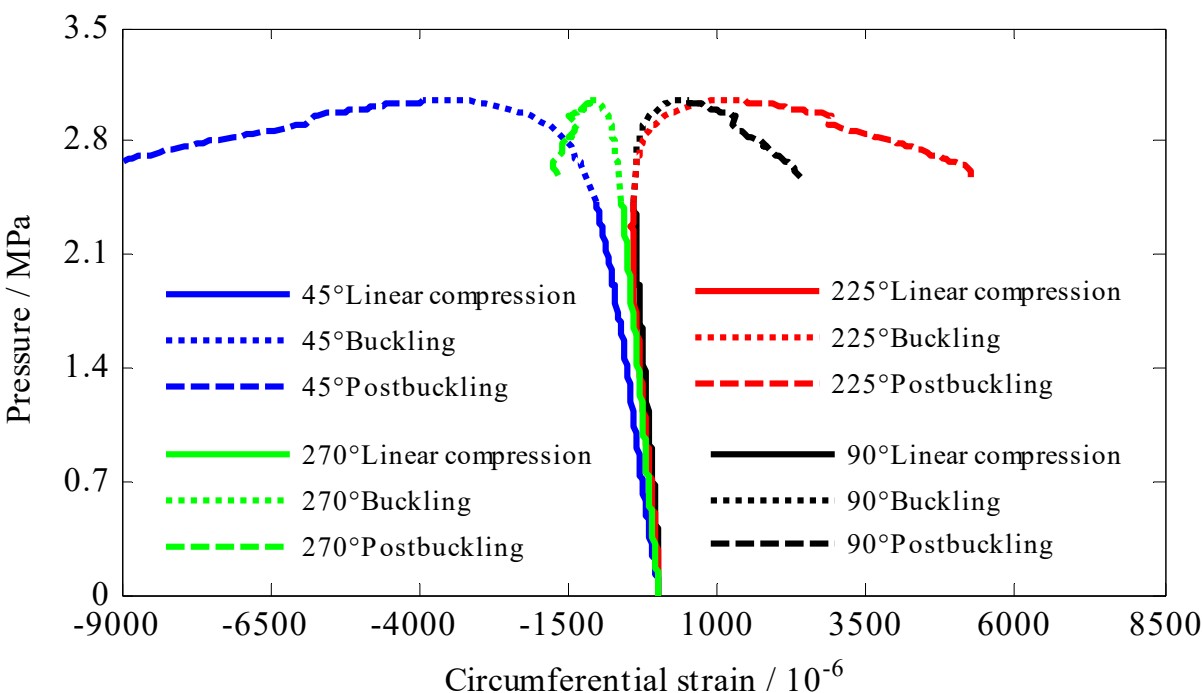

**Figure 10.** Circumferential strain at the boundary of crest and trough of shell A#.

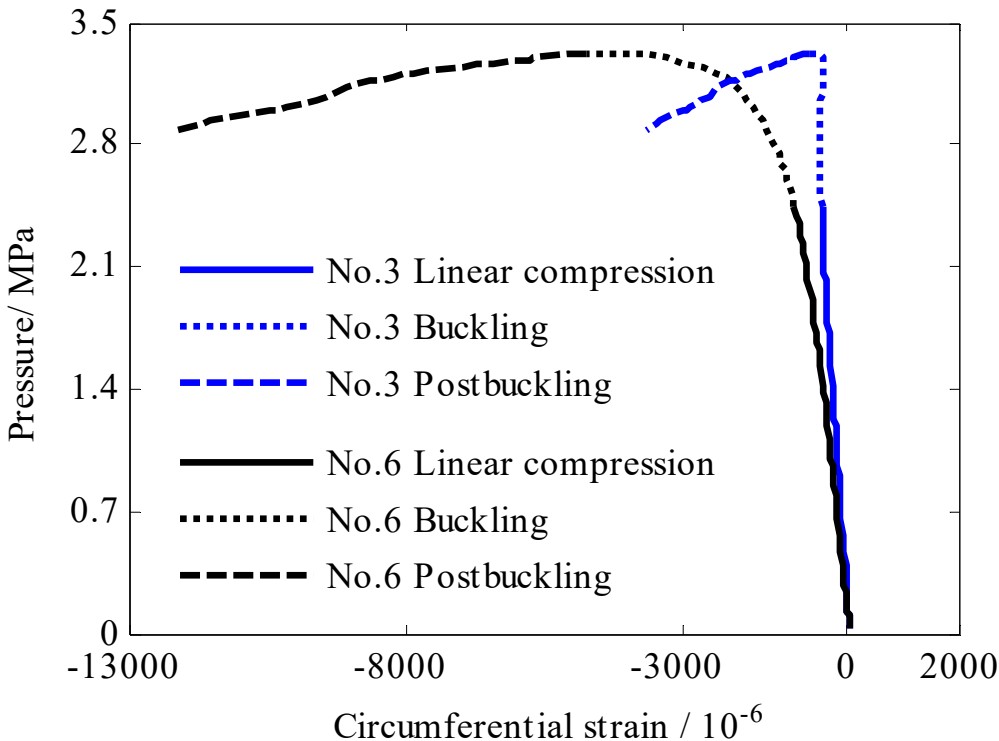

**Figure 11.** Circumferential strain at the crest of the buckling wave of shell C#.

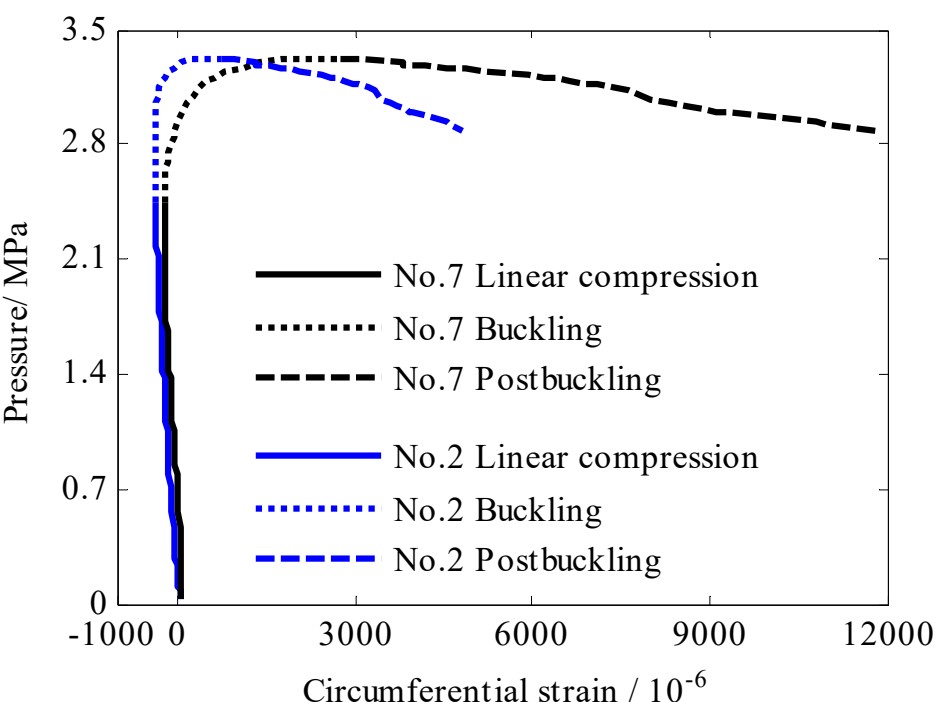

**Figure 12.** Circumferential strain at the trough of the buckling wave of shell C#.

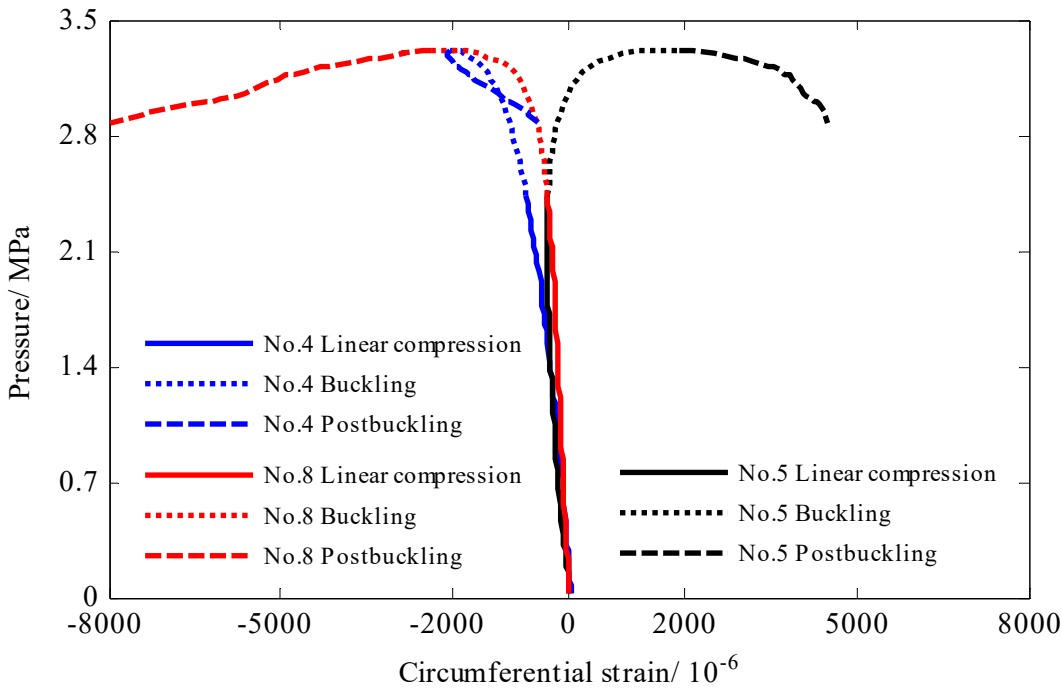

**Figure 13.** Circumferential strain at the boundary of the crest and trough of shell C#.

As for the perforated cylinder C#, in the linear compression stage 0–2.44 MPa circumferential strain presented linearly compressive and increased linearly with the hydrostatic pressure. In buckling stage 2.44–3.32 MPa, strain magnitude tended to be nonlinearly varying with hydrostatic pressure. Additionally, two phenomena, which also manifested on the perfect cylinder A#, occurred. On the one hand, circumferential strain of the measuring points No.3 and No.6 located at the crest (see Figure 11) of the buckling wave showed compressive in the buckling stage. On the other hand, strain reversal of No.2 and No.7 also occurred at the trough (Figure 12) in this stage. At the boundary of crest and trough

of the buckling wave (see Figure 13), strain gage No.8 presented compressive strain in load history, and strain gage No.5 exhibited reversal phenomenon in the buckling stage. With regard to No.4, the magnitude of compressive strain decreases sharply when the deformation developed into postbuckling stage, which might be cause by material damage at this location. Focus on the influence of reinforcement part on the strain response of the perforated composite cylindrical shell. We review strain response of No.2 (see Figure 12) and No.8 (see Figure 13) located in the reinforcement boundary area. Strain magnitude of these two points was in good agreement with other measuring points in linear compression stage. This phenomenon indicated that the reinforcement design could ensure the continuity of the hoop stiffness which also can be verified by the fact that both the terminated buckling load $p_{\text{terminate}}$ and the final failure load $p_{\text{failure}}$ of the perforated shell B# and C# were higher than that of perfect shell A#.

### 4. Conclusions

In this study, a test platform for real-time observation of shell deformation under hydrostatic pressure was built. The buckling and post-buckling behavior of perfect and perforated composite cylindrical shell subjected to external hydrostatic pressure was investigated. The main conclusions were summarized as follows.

(1) The buckling behavior of composite cylindrical shell under hydrostatic pressure was a progressive evolution process which accounted for more than 20% of the load history, and the load carrying capacity was still increasing in this stage.

(2) As for the post-buckling behavior, the initial buckling wave was extended while the load carrying capacity of the shell decreases. Both the perfect and perforated composite cylindrical shell collapsed at the trough of the buckling wave.

(3) The buckling shape of the perfect composite cylindrical shell was uniformly symmetrically distributed along the circumference. As for the perforated composite cylindrical shell, the buckling wave presented non-uniform and asymmetrical distribution due to the non-uniform circumferential stiffness.

(4) Regarding the failure mode of thin-walled composite cylindrical shell under hydrostatic pressure, it presented buckling failure initially. As the shell deformation increased, material failure occurred subsequently, and the shell collapsed eventually.

**Author Contributions:** Validation, L.-L.J. and Z.-Q.Y.; writing—original draft preparation, K.-C.S.; supervision, G.P. All authors have read and agreed to the published version of the manuscript.

**Funding:** This research was funded by the National Natural Science Foundation of China (52101376), the Fundamental Research Funds for the Central Universities (3102019JC006).

**Institutional Review Board Statement:** Ethical review and approval were waived for this study not involving humans or animals.

**Informed Consent Statement:** Not applicable.

**Data Availability Statement:** Not applicable.

**Conflicts of Interest:** The authors declare no conflict of interest.

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
