# Peer review of "Buckling and Post-Buckling Behavior of Perfect/Perforated Composite Cylindrical Shells under Hydrostatic Pressure"

_jmse, doi:10.3390/jmse10020278_

Round 1

Reviewer 1 Report

The authors have investigated the buckling response of thin composite hulls experimentally. In my opinion there is not much novelty in the current work. The authors claim that (1) previous studies were done in closed chambers with no visual results of the buckle progress, (2) previous test were conducted on closed hulls. Well, I think both claims are questionable. Moreover, there are missing information such as: (i) clear explanation of the manufacturing process, (ii) details of samples A#,B#,C# and (iii) rational for selecting these samples. The finite element model is not defined. It's not clear what type of material, analyses, sensitivity, failure model,... are incoporated in the FEA. 

Discussion of results lack the physical interpretation, comparison with theory (analytical solutions) and is basically images and pressure vs. strains. I don't see how these can benefit readers of JMSE.

I have not rejected the paper, but expect to see a significantly revises re-submission if authors care to consider.

Author Response

Reviewer 1

The authors have investigated the buckling response of thin composite hulls experimentally. In my opinion there is not much novelty in the current work. I have not rejected the paper, but expect to see a significantly revises re-submission if authors care to consider.

1 The authors claim that (1) previous studies were done in closed chambers with no visual results of the buckle progress, (2) previous test were conducted on closed hulls. Well, I think both claims are questionable.

Response: Thank you for your valuable suggestion. In the revised version, the authors have reviewed several experimental investigations such as follow:

Carvelli V, Panzeri N, Poggi C. Buckling strength of GFRP under-water vehicles. Composites Part B Engineering, 2001, 32:89-101.

Hur S H, Son H J, Kweon J H, et al. Postbuckling of composite cylinders under external hydrostatic pressure. Composite Structures, 2008, 86:114-124.

Moon C J, Kim I H, Choi B H. Buckling of filament-wound composite cylinders subjected to hydrostatic pressure for underwater vehicle applications. Composite Structures, 2010, 92:2241-2251.

In the above references, the final buckling failure or material failure were presented and discussed. However, the buckling evolution of composite cylindrical shells under hydrostatic pressure has not been studied, especially by experimental method.

2 Moreover, there are missing information such as: (i) clear explanation of the manufacturing process, (ii) details of samples A#,B#,C# and (iii) rational for selecting these samples.

Response: Thank you for pointing this out. In the revised version, the author has added detail explanation of the manufacturing process of the shell, including the perforate process and reinforcement design. The geometric model and dimensions of the samples were described in detail. Additionally, the samples A#, B# and C# were cut from a perfect cylinder which was formed on one core mold and experienced curing and demoulding. The cylinder was cut into three cylinders with equal length, and two of them were perforated and reinforced. Therefore, the authors believe that the selection of samples is rational.

3 The finite element model is not defined. It's not clear what type of material, analyses, sensitivity, failure model,... are incoporated in the FEA.

Response: Thank you for pointing this out. In the revised version, the simulation has been presented clearly, and the details of structure modeling, type of element, material, boundary conditions, mesh discretization and analysis process have been discussed.

4 Discussion of results lack the physical interpretation, comparison with theory (analytical solutions) and is basically images and pressure vs. strains. I don't see how these can benefit readers of JMSE.

Response: Thank you for pointing this out. In the revised version, more interpretation has been provided in the results and discussion part. To be honest, this paper aimed to reveal the instability evolution process of the composite cylindrical shell under hydrostatic pressure based on experimental observations, and maybe readers benefit less from this article in terms of theoretical innovations. However, the authors have contributions in experimental platform construction that can provide researchers with important design references. In future studies, the authors will try their best to use analytical solution to physically explain the progressive buckling process.

Reviewer 2 Report

This study has been devoted to experimentally investigate the buckling and post-buckling behavior of perfect and perforated cylindrical shells subjected to hydrostatic pressure. An applicable experimental platform has been proposed in this research. The topic of the research is interesting. However, a major revision is required for this paper:

1- The highlights of the research should be presented. The authors should clarify why it is necessary to use experimental study for this issue while there are several numerical analyses in which this issue can be investigated completely.

2- In the abstract, the authors should discuss the type of materials used in this model.

3- The literature review on the numerical studies should be improved. There are several numerical studies in which the nonlinear behavior of cylindrical shells has been investigated such as follow: AIAA journal 8.2 (1970): 229-235.-Journal of pressure vessel technology 131.6 (2009)-Journal of the Brazilian Society of Mechanical Sciences and Engineering 41 (10), 419, 2019-Mechanics of Advanced Materials and Structures 21.6 (2014): 490-504.-World Journal of Engineering 16 (5), 636-647, 2019-International Journal of pressure vessels and piping 79.5 (2002): 351-359.

4- The FEM simulation should be presented clearly. The details of structure modeling in the FEM software and mesh discretization should be discussed. 

5- The explanations provided for figures 7 to 12 are not sufficient. More interpretation should be provided for these figures. 

Author Response

Reviewer 2

This study has been devoted to experimentally investigate the buckling and post-buckling behavior of perfect and perforated cylindrical shells subjected to hydrostatic pressure. An applicable experimental platform has been proposed in this research. The topic of the research is interesting. However, a major revision is required for this paper:

1- The highlights of the research should be presented.

Response: Thank you for pointing this out. The highlights of the research were as following.

1) A test platform was established that allows researchers to observe the deformation of composite cylindrical shells under hydrostatic pressure in real time. 2) The buckling behavior was a progressive evolution process which accounted for 20% of the load history, and strain reversal phenomenon generally occurred at the trough of the buckling wave. 3) In the postbuckling stage, the load carrying capacity of the shell gradually decreased while the magnitude of strain continued increasing. All the composite cylindrical shells collapsed at the trough of the buckling wave. 4) It was validated the reinforcement design could effectively ensure the load carrying capacity of the perforated composite cylindrical shell.

2-The authors should clarify why it is necessary to use experimental study for this issue while there are several numerical analyses in which this issue can be investigated completely.

Response: Thank you for your valuable suggestion. From the review of the previous published literature studies, the researchers generally obtained the critical buckling pressure directly from analytical solution and numerical method. In some researches regarding to external hydrostatic pressure tests, it is found that nonlinear strain behavior occurred before the shell collapsed. The collapse pressure was defined as the critical buckling pressure but the nonlinear strain behavior was not explained clearly. Therefore, it is necessary to distinguish and reveal differences in nonlinear strain behavior and buckling phenomena through experimental study.

3- In the abstract, the authors should discuss the type of materials used in this model

Response: Thank you for your valuable suggestion. In the revised manuscript, the material of samples has been explained in detail.

4- The literature review on the numerical studies should be improved. There are several numerical studies in which the nonlinear behavior of cylindrical shells has been investigated such as follow:

-AIAA journal 8.2 (1970): 229-235.

-Journal of pressure vessel technology 131.6 (2009).

-Journal of the Brazilian Society of Mechanical Sciences and Engineering 41 (10), 419, 2019.

-Mechanics of Advanced Materials and Structures 21.6 (2014): 490-504.

-World Journal of Engineering 16 (5), 636-647, 2019.

-International Journal of pressure vessels and piping 79.5 (2002): 351-359.

Response: Thank you for your valuable suggestion. In the revised manuscript, the authors have reviewed studies on the nonlinear behavior of cylindrical shells. Related researches have been cited in this work.

5- The FEM simulation should be presented clearly. The details of structure modeling in the FEM software and mesh discretization should be discussed.

Response: Thank you for pointing this out. In the revised version, the simulation has been presented clearly, and the details of structure modeling, type of element, material, boundary conditions, mesh discretization and analysis process have been discussed.

6- The explanations provided for figures 7 to 12 are not sufficient. More interpretation should be provided for these figures.

Response: Thank you for your valuable suggestion. In the revised version, the authors present sufficient explanations for Figures 7-12.

Round 2

Reviewer 1 Report

The authors have addressed my comments to satisfaction. The article can be published in JMSE.

Reviewer 2 Report

The authors revised the article completely according to the comments.